# The impact of role overload on healthcare workers' psychological distress and well-being: The mediating role of task distraction and moderating roles of resilience and emotional intelligence

Faseeh Iqbal[1], Fatima Noreen[2], Sami Iqbal[3], Umer Farooq[4], Quratulain Batool[1], Sumbal Shahbaz[5]*

1 MS Healthcare Management, Faculty of Management Sciences, Riphah International University, Islamabad, Pakistan, 2 Department of Health Professional Technologies, The University of Lahore, Lahore, Pakistan, 3 BS Public Health, Khyber Medical University, Peshawar, Pakistan, 4 Isra University, Al Nafees Medical College & Hospital, Islamabad, Pakistan, 5 Department of Health Professional Technologies, The University of Lahore, Lahore, Pakistan

* sumbal.shahbaz@dhpt.uol.edu.pk

## Abstract

Healthcare professionals, including physicians, nurses, and allied health workers, often balance patient care demands under high workloads and limited resources. Maintaining workforce well-being is essential for healthcare system effectiveness. This study examined the impact of role overload (RO) on psychological distress (PD) and psychological well-being (PWB), focusing on the mediating role of task distraction (TD) and the moderating roles of employee resilience (ER) and emotional intelligence (EI). A cross-sectional design was employed, and data were collected from 600 healthcare workers in public and private hospitals in Lahore, Islamabad, Peshawar, Swabi, and Mardan, Pakistan. Validated psychometric scales were used to measure RO, ER, TD, PD, PWB, and EI. The results showed that RO was positively associated with PD and negatively associated with PWB. TD partially mediated both associations. ER moderated the relationship between RO and TD, while EI moderated the associations between TD and both PD and PWB. These findings indicate that higher resilience and emotional intelligence buffer the negative impact of workload and distraction on mental health outcomes. This study highlights the importance of organizational interventions, such as resilience training, emotional intelligence development, and strategies to reduce task distraction, to strengthen healthcare workforce well-being and improve patient care quality.

**Data availability statement:** The data is available within the manuscript and its Supporting Information files.

**Funding:** The authors received no specific funding for this work.

**Competing interests:** The authors have declared that no competing interests exist.

## Introduction

Healthcare professionals including physicians, nurses, and allied health workers constantly balance the delivery of patient care under high workloads and scarce resources. An engaged workforce, characterized by vigor, dedication, and absorption, is crucial for healthcare system effectiveness [1]. Unfortunately, many healthcare workers experience role overload, where workload exceeds their capacity, leading to psychological distress, including stress, anxiety, burnout, and emotional exhaustion [2–4]. This issue, exacerbated by long working hours and emotional demands, is prevalent not just in Pakistan but globally, with post-pandemic pressures intensifying existing stressors [5]. Research indicates that emotional intelligence (EI) and resilience are key in mitigating the psychological effects of role overload, enhancing stress management and overall well-being in healthcare workers [6–8]. Training in these areas has proven effective in supporting mental health, especially in systems where understaffing and emotional burden are significant challenges [9]. Role overload, defined by rapid work pace, high workloads, and time pressure, negatively impacts healthcare workers' well-being and the quality of care, leading to medical errors, staff attrition, and reduced patient satisfaction [1,10]. Reducing role overload is essential for maintaining an optimal healthcare workforce and ensuring high-quality care [11]. Psychological distress, such as anxiety and burnout, reduces job satisfaction and performance, increasing turnover among healthcare workers. In contrast, psychological well-being enhances mental health, job satisfaction, and patient care quality [12,13]. Overload heightens the risk of burnout and distress, leading to reduced productivity, higher costs, absenteeism, and lower-quality care [14].

Despite the well-documented impact of role overload, limited research exists on mediators and moderators in this relationship. Task distractions such as interruptions and competing tasks exacerbate stress and impair job performance, worsening the effects of role overload and psychological distress over time [15–17]. Resilience, defined as the capacity to withstand chronic stress while maintaining psychological health, is vital in helping healthcare workers manage role overload and prevent burnout [18,19].

The role of Emotional Intelligence (EI), the ability to perceive, regulate, and use emotions in alleviating distress from role overload has often been neglected. High-EI employees report lower burnout, better stress management during crises, and enhanced coping abilities in high-demand situations [20–22]. EI also helps mitigate post-pandemic trauma in healthcare workers, promoting adaptive emotional regulation [23], and moderating the relationship between job demands and anxiety [24]. Organizational strategies like AI-driven emotional intelligence (EI) workshops are addressing role overload, with cross-cultural studies highlighting EI's role in stress management in healthcare [8,25]. However, the combined effect of EI and resilience on psychological strain remains unclear. This study proposes a new moderating framework integrating EI and resilience, supported by transformative AI interventions, to improve well-being and reduce distress in post-pandemic healthcare settings [26].

The impact of role overload on healthcare workers' psychological distress and well-being can be comprehended through various frameworks. The Job

Demands-Resources Model suggests that role overload depletes resources, leading to burnout, while resilience and emotional intelligence (EI) act as buffers. The Transactional Model of Stress and Coping highlights how EI and task shielding help mitigate stress, while the Conservation of Resources (COR) Theory links resource depletion and emotional burnout, emphasizing resilience as a protective factor. Resilience Theory asserts that individuals with greater resilience maintain well-being in the face of challenges.

This study explores the relationship between role overload, task distraction, employee resilience, and EI, examining their impact on psychological distress and well-being in healthcare workers. It investigates how task distraction influence's role overload's psychological outcomes, with resilience and EI as moderators. The findings will help healthcare institutions implement better organizational support to reduce the negative effects of role overload, improving both worker well-being and patient care outcomes. Healthcare workers in Pakistan often report high role overload in under-resourced settings. Prior studies have linked overload with distress and lower well-being, but the role of task distraction as a statistical mediator remains under-examined in healthcare samples. Evidence on whether resilience and emotional intelligence moderate these associations is also mixed, and large, multi-site data from Pakistani hospitals are limited [27]. This study addresses these gaps by testing whether task distraction statistically mediates the associations between overload and distress and between overload and well-being, and whether resilience and emotional intelligence moderate these associations. Unlike most global studies, this research draws on multi-site data from Pakistan, providing context-specific insights into how workplace stressors and personal resources interact in low-resource healthcare systems.

## Materials and methods

This study used a cross-sectional design to investigate the association among role overload, task distraction, employee's resilience, psychological stress, and psychological well-being. Data were collected from 600 healthcare workers (Physician, Nurses, Allied Health Professionals) working in both public and private healthcare sectors. The diverse sample strengthens the generalizability of the findings across multiple institutional and geographic settings in Pakistan, with participant drawn from both public and private hospitals in Lahore, Islamabad, Peshawar, Mardan, and Swabi. This study adopted a quantitative approach using a purposive sampling technique to collect data from healthcare workers. In order to capture a representative sample of healthcare workers, data were collected over a time span from 16 Dec 2024–28 Feb 2025, through online surveys as well as by personally distributing questionnaires at the hospitals, which increases the study's generalizability and applicability to various healthcare contexts.

The researchers engaged the eligible participants, elucidating the study's objectives and advantages, as well as the participants' right to withdraw from the study. A verbal informed consent was then acquired from those who wanted to take part in the study. Many individuals, especially healthcare professionals, often exhibit reluctance to provide written consent due to apprehension or distrust. To ensure confidentiality and anonymity, the participants were told to put the questionnaire in the enclosed envelope, seal it, and give it to the researchers. Study participation was entirely voluntary. The freedom to leave the study at any moment without facing any repercussions was communicated to the participants. Participants were also given the assurance that their personal data would be kept confidential.

"The study utilized a 5-point Likert scale, with response options ranging from 1 = (Strongly Disagree) to 5 = (Strongly Agree) to measure employee resilience, role overload, task distraction, psychological distress, and psychological well-being. Role overload was measured using the 5-item scale developed by Peterson and originally reported in a 21-nation study on *Role Conflict, Ambiguity, and Overload* [28]. Employee resilience was assessed with a 6-item scale developed by Smith [29] Distraction from task focus was measured using a 4-item scale developed by Seddigh [30]. The Psychological stress scale was based on Kessler [31] which contains 6 items. Psychological well-being on the other hand was evaluated using six items from Reker and Wong [32] Emotional Intelligence, Wong and Law [33] formulated the self-assessment questionnaire known as "The Wong and Law Emotional Intelligence Scale (WLEIS)" in 2002. The WLEIS is composed of 16 items structured into four categories: self-emotion appraisal (SEA), others' emotion appraisal (OEA), use of emotion

(UOE), and regulation of emotion (ROE). Each section contains four items which are responded by using a Likert scale of 1–7, with 1 representing 'strongly disagree' and 7 representing 'strongly agree." This scale is known to have strong psychometric properties across cultures including reliability estimates (Cronbach's alpha) of 0.83 to 0.92, which makes it appropriate to measure EI among healthcare practitioners.

The data were analyzed with the Statistical Package for Social Sciences (SPSS). The respondents' demographic information was outlined using descriptive statistics. An assessment of reliability was performed to test the internal consistency for the constructs of the instrument used in the research. Furthermore, correlation analyses together with regression analyses were performed to examine the relationships among role overload, task distraction, employee resilience, psychological stress, and psychological well-being. To test the proposed model and examine the mediating and moderating impacts, mediation and moderation analyses were done using Hayes' PROCESS macro. This advanced technique captures all of the interrelationships among the study variables in a single framework, ensuring the results are helpful to understanding how healthcare systems can mitigate role overload while enhancing the healthcare workers' psychological wellness.

The study was conducted in accordance with the ethical standards of the institutional research committee. Ethical approval was obtained from the Institutional Review Board (IRB) Ref No: REC-UOL-/529/08/24 from University of Lahore. Verbal informed consent was obtained from participants during in-person data collection at hospitals, while online informed consent was obtained for participants completing surveys electronically. This dual approach ensured that all participants provided informed consent prior to data collection. All procedures involving human participants were conducted in accordance with institutional guidelines and the ethical principles outlined in the 1964 Declaration of Helsinki and its subsequent revisions.

## Theoretical Framework

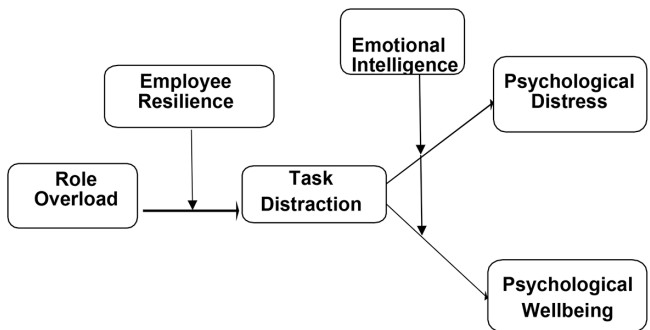

IV, Role Overload, DV. Psychological Distress, DV. Psychological Wellbeing, MED. Task Distraction, MOD. Employee Resilience, MOD. Emotional Intelligence.

## Results

Data were obtained from a diverse sample of 600 healthcare professionals, including 162 physicians (27%), 240 nurses (40%), and 198 allied health professionals (33%), across public and private hospitals in Lahore, Islamabad, Peshawar, Mardan, and Swabi (Fig 1). The sample comprised 390 males (65%) and 210 females (35%), reflecting a gender distribution consistent with regional workforce trends (Fig 2). Participants represented varying levels of professional experience: 132 (22%) had over 10 years of experience, 222 (37%) had approximately 5 years, and 246 (41%) were early-career professionals with 1–2 years of experience (Fig 3). The inclusion of healthcare employees from multiple cities and institutional settings strengthens the generalizability of the study's findings across Pakistan's diverse healthcare landscape.

This demographic distribution provided a robust basis for examining the impact of role overload, task distraction, emotional intelligence, and resilience across different career stages. Data analysis was conducted using IBM SPSS Statistics

PLOS Mental Health

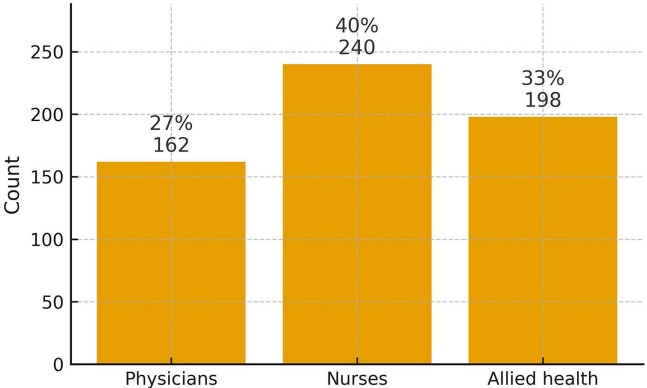

**Fig 1. Distribution of Healthcare workers (N = 600).**

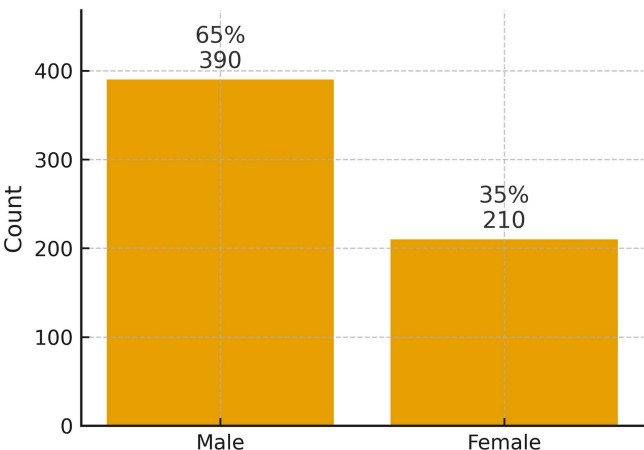

**Fig 2. Gender Distribution of Healthcare workers (N = 600).**

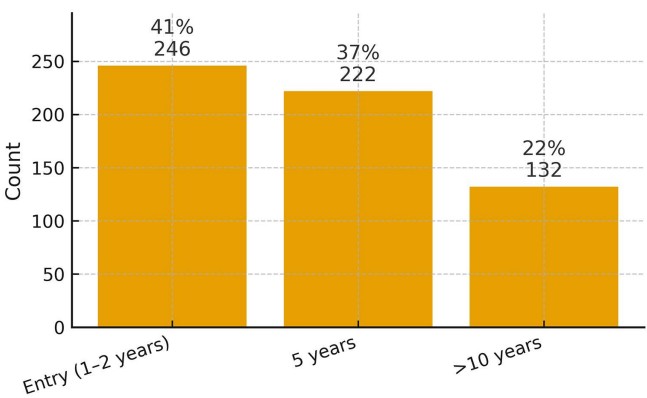

**Fig 3. Experiences of Healthcare workers (N = 600).**

Version 26, including reliability testing, correlation analysis, multiple regression, and mediation and moderation analyses to explore the hypothesized relationships.

Fig 1 shows the professional composition (N = 600): 162 physicians (27%), 240 nurses (40%), and 198 allied health professionals (33%), drawn from public and private hospitals in Lahore, Islamabad, Peshawar, Mardan, and Swabi.

Fig 2 shows the gender distribution of healthcare workers (N = 600): 390 males (65%) and 210 females (35%).

Fig 3 shows that participants varied in terms of professional experience: 132 individuals (22%) had over 10 years of experience, 222 (37%) had approximately 5 years of experience, and 246 (41%) were entry-level staff with 1–2 years of experience.

Table 1 presents the Cronbach's Alpha coefficients assessing the internal consistency of the study variables. All scales demonstrated acceptable to high reliability. Role Overload (RO) showed strong reliability (α = 0.891), Employee Resilience (ER) had acceptable reliability (α = 0.790), and Task Distraction (TD) demonstrated good reliability (α = 0.878). Psychological Distress (PD) showed satisfactory reliability (α = 0.850), while Psychological Well-Being (PWB) also displayed strong reliability (α = 0.890). These findings confirm that the measurement scales used in this study were internally consistent and appropriate for assessing the respective constructs.

Table 2 presents the correlation coefficients among Role overload (RO), Employee Resilience (ER), Task Distraction (TD), Psychological Distress (PD), Psychological Wellbeing (PWB), RO show strong positive association with TD (r = 0.764) and PD (r = 0.688). PWB demonstrated negative associations with RO (r = −0.572), TD (r = −0.511), and PD (r = −0.545). ER showed weak positive associations with RO (r = 0.164) and TD (r = 0.168) and a weak negative association with PWB (r = −0.142).

Table 3 presents that Role Overload (β = 0.560, p < 0.001) was positively associated with Psychological Distress (PD). Role Overload (β = -0.643, p < 0.001) was negatively associated with Psychological Well-Being (PWB). Role Overload (β = 0.450, p < 0.001) was positively associated with Task Distraction (TD). Task Distraction (β = 0.448, p < 0.001) was positively associated with Psychological Distress (PD). Task Distraction (β = -0.358, p < 0.001) was negatively associated with Psychological Well-Being (PWB). β (Beta Coefficient) shows how strong and whether the correlational relationships that exist between the predictors and the outcomes is positive or negative.

**Table 1. Cronbach's Alpha.**

| Variable Name | No of Items | Cronbach's Alpha |
|---|---|---|
| 1. RO | 5 | .891 |
| 2. ER | 6 | .790 |
| 3. TD | 4 | .878 |
| 4. PD | 6 | .850 |
| 5. PWB | 6 | .890 |

**Table 2. Correlation Analysis.**

| | 1 | 2 | 3 | 4 | |
|---|---|---|---|---|---|
| RO | 1.000 | | | | |
| ER | 0.164 | 1.000 | | | |
| TD | 0.764 | 0.168 | 1.000 | | |
| PD | 0.688 | 0.094 | 0.712 | 1.000 | |
| PWD | -0.572 | -0.142 | -0.511 | -0.545 | 1.000 |

**Table 3. Regression Analysis–Direct Relationships.**

| Predictors | | | | |
|---|---|---|---|---|
| | β | R² | R | p-value |
| RO→PD | 0.560 | 0.630 | 0.794 | <0.001 |
| RO→PWB | -0.643 | 0.645 | 0.803 | <0.001 |
| RO→TD | 0.450 | 0.632 | 0.796 | <0.001 |
| TD→PD | 0.448 | 0.635 | 0.799 | <0.001 |
| TD→PWB | -0.358 | 0.645 | 0.803 | <0.001 |

R² is included to show variability explained by the dependent independent factors. R gives the strength of the linear relationship between each pair of variables, p-value confirms whether the relationship is statistically significant (values < 0.001 are highly significant).

As shown in Table 4, the association between Role Overload (RO) and Psychological Distress (PD) through Task Distraction (TD) included a direct effect (β = 0.448, p < 0.001) and an indirect effect (β = 0.201, 95% CI [0.112, 0.300]), indicating partial mediation. For Psychological Well-Being (PWB), the indirect effect was negative (β = −0.160, 95% CI [−0.278, −0.042]), suggesting that Task Distraction partially explained the association between Role Overload and reduced well-being. The negative indirect effect for PWB emphasizes the harmful role of Task Distraction in diminishing psychological well-being as Role Overload increases. Overall, these findings highlight the complex nature of the relationship between Role Overload and mental health, with Task Distraction acting as a partial mediator in both negative and positive pathways. The VAF (Variance Accounted For) indicates the proportion of the total effect explained by the mediator. For the pathway RO→TD→PD, the VAF of 20.1% shows that Task Distraction (TD) accounts for 20.1% of the effect of Role Overload (RO) on Psychological Distress (PD). Similarly, the VAF of 16% for RO→TD→PWB demonstrates that TD mediates 16% of the effect on Psychological Well-Being (PWB). These results highlight TD as a partial mediator in both pathways.

Table 5 presents that Employee Resilience moderated the association between Role Overload and Task Distraction (β = −0.198, p = 0.031, 95% CI [−0.371, −0.025]). Emotional Intelligence moderated the associations between Task Distraction

**Table 4. Mediation Regression Analysis.**

| Pathway | Effect Type | β | SE | t | p | 95% CI (Bootstrap) | VAF | Interpretation |
|---|---|---|---|---|---|---|---|---|
| **RO→TD→PD** | Total Effect (c) | 0.649 | 0.078 | 8.32 | <0.001 | [0.498, 0.799] | 20.1% | Overall association between RO and PD |
| | Direct Effect (c′) | 0.448 | 0.072 | 6.22 | <0.001 | [0.306, 0.590] | – | Association controlling for TD |
| | Indirect Effect | 0.201 | – | – | <0.001 | [0.112, 0.300] | 20.1% | Mediation via TD (partial mediation) |
| **RO→TD→PWB** | Total Effect (c) | 0.285 | 0.091 | 3.13 | 0.002 | [0.106, 0.464] | 16% | Overall association between RO and PWB |
| | Direct Effect (c′) | 0.445 | 0.080 | 5.56 | <0.001 | [0.287, 0.603] | – | Association controlling for TD |
| | Indirect Effect | −0.160 | – | – | 0.015 | [−0.278, −0.042] | 16% | Negative mediation via TD (partial mediation) |

**Table 5. Moderation Regression Analysis.**

| Interaction Term | β | SE | t | p | 95% CI (Bootstrap) | Interpretation |
|---|---|---|---|---|---|---|
| RO×ER→TD | −0.198 | 0.089 | −2.223 | 0.031 | [−0.371, −0.025] | ER moderates the RO→TD association, with higher resilience weakening the impact of RO on TD. |
| TD×EI→PD | −0.211 | 0.076 | −2.776 | 0.006 | [−0.360, −0.062] | EI moderates the TD→PD relationship, with higher EI buffering the effect of task distraction on PD |
| TD×EI→PWB | 0.235 | 0.082 | 2.865 | 0.005 | [0.073, 0.397] | EI moderates the TD→PWB relationship, with higher EI mitigating the negative impact of TD on PWB |

and both Psychological Distress (β = −0.211, p = 0.006, 95% CI [−0.360, −0.062]) and Psychological Well-Being (β = 0.235, p = 0.005, 95% CI [0.073, 0.397]). These results suggest that higher resilience and emotional intelligence attenuate negative relationships among study variables. The findings emphasize the importance of fostering resilience and emotional intelligence to mitigate the impact of role overload and task distractions on mental health

**Key findings and interpretation**

RO × ER → TD: The interaction between Role Overload and Employee Resilience was statistically significant (β = −0.198, p = 0.031), indicating that resilience moderated the association between role overload and task distraction. Higher resilience levels were linked to a weaker positive association between role overload and task distraction, although the relationship was not completely eliminated.

TD × EI → PD: Emotional Intelligence significantly moderated the association between task distraction and psychological distress (β = −0.211, p = 0.006). The negative coefficient suggests that higher emotional intelligence levels were associated with a weaker positive relationship between task distraction and psychological distress, indicating a buffering role for emotional intelligence in high-stress environments.

TD × EI → PWB: Emotional Intelligence also moderated the association between task distraction and psychological well-being (β = 0.235, p = 0.005). Higher emotional intelligence levels were linked to a weaker negative association between task distraction and psychological well-being, suggesting that emotional intelligence helps protect psychological well-being even when task distractions are present.

Table 6 presents the summary of hypothesis.

## Discussion

Role overload was positively associated with psychological distress and negatively associated with psychological well-being. Task distraction significantly mediated both associations, while resilience and emotional intelligence moderated these relationships. These findings align with existing role stressor research in healthcare and extend prior work by quantifying how distraction relates to overload outcome associations in a multi-site Pakistani sample. The observed moderation by resilience and emotional intelligence supports resource-based perspectives, indicating that personal resources are linked to variability in these associations across staff.

This study examined the associations between Role Overload (RO), Psychological Distress (PD), and Psychological Well-Being (PWB) among healthcare professionals, with Task Distraction (TD) as a mediator and Employee Resilience (ER) and Emotional Intelligence (EI) as moderators. The results demonstrated that RO was positively associated with PD and negatively associated with PWB, aligning with research linking excessive workload to higher stress, burnout, and poorer mental health outcomes, particularly during the COVID 19 pandemic [34,35].

The finding that RO was positively associated with PD (β = 0.560, p < 0.001) corroborates studies identifying high workload demands as key factors associated with emotional fatigue and stress among healthcare workers [11,36,37]. As noted in prior research, prolonged work overload is linked with burnout and emotional exhaustion, emphasizing the need for systemic workload reforms. Similarly, the negative association between RO and PWB (β = -0.643, p < 0.001) is consistent with studies suggesting that prolonged stress depletes psychological resources, contributing to disengagement and reduced well-being [38–40]. Our finding that RO was positively associated with TD (β = 0.450, p < 0.001) is also supported by research highlighting the cognitive burden of task interruptions in high-pressure environments and their association with reduced focus and increased errors [39,40].

TD was positively associated with PD (β = 0.448, p < 0.001) and negatively associated with PWB (β = -0.358, p < 0.001), suggesting that task distractions are linked with greater emotional and cognitive strain and reduced psychological

**Table 6. Summary of Hypothesis and Results.**

| Hypothesis | Statement | Results |
|---|---|---|
| H1a | Role Overload (RO) significantly positively associated Psychological Distress (PD) | Statistically significant ($\beta = 0.560$, $p < 0.001$) |
| H1b | Role Overload (RO) significantly negatively associated Psychological Well-Being (PWB) | Statistically significant ($\beta = -0.643$, $p < 0.001$) |
| H2 | Role Overload (RO) significantly positively associated Task Distraction (TD) | Statistically significant ($\beta = 0.450$, $p < 0.001$) |
| H3a | Task Distraction (TD) significantly positively associated Psychological Distress (PD) | Statistically significant ($\beta = 0.448$, $p < 0.001$) |
| H3b | Task Distraction (TD) significantly negatively associated Psychological Well-Being (PWB) | Statistically significant ($\beta = -0.358$, $p < 0.001$) |
| H4a | Task Distraction (TD) significantly mediates the Association between Role Overload (RO) and Psychological Distress (PD). | Statistically significant (Indirect Effect $\beta = 0.201$, 95% CI [0.112, 0.300], $p < 0.001$) |
| H4b | Task Distraction (TD) significantly mediates the Association between Role Overload (RO) and Psychological Well-Being (PWB). | Statistically significant (Indirect Effect $\beta = -0.160$, 95% CI [−0.278, −0.042], $p = 0.015$) |
| H5 H6 H7 | Employee Resilience (ER) moderates the Association between Role Overload (RO) and Task Distraction (TD), such that higher resilience weakens the positive effect of role overload on task distraction. (TD) Emotional Intelligence moderates the Association between Task Distraction (TD) and Psychological Distress (PD), such that higher EI attenuates the positively effect of task distraction on distress. Emotional Intelligence (EI) moderates the Association between Task Distraction (TD) and Psychological Well-Being (PWB), such that higher EI mitigates the negatively effect of task distraction on well-being. | Statistically significant (ER × RO → TD $\beta = -0.198$, $p = 0.031$) Statistically significant (EI × TD → PD $\beta = -0.211$, $p = 0.006$) Statistically significant (EI × TD → PWB $\beta = 0.235$, $p = 0.005$) |

well-being [41,42]. Moreover, the mediation of RO's associations with PD (indirect effect = 0.201; VAF = 0.310) and PWB (indirect effect = -0.160; VAF = 0.561) by TD aligns with cognitive load theory, which posits that cognitive resources become depleted under overwhelming demands, impairing mental and emotional functioning [38,43]. These findings highlight the role of task distractions in shaping the associations between workload and mental health outcomes.

The moderating effects of ER and EI further indicate how personal resources are associated with reduced vulnerability to workload-related stress. The negative moderation of ER on the RO-TD association ($\beta = -0.198$, $p = 0.031$) suggests that individuals with higher resilience demonstrate weaker associations between workload demands and distractions, although this effect does not completely diminish the influence of RO [44–46]. Similarly, the moderation of EI in the TD–PD ($\beta = -0.211$, $p = 0.006$) and TD–PWB ($\beta = 0.235$, $p = 0.005$) associations highlights the role of emotional regulation in mitigating stress in healthcare environments. High-EI individuals demonstrate stronger emotional regulation capacities, which are associated with reduced psychological strain and enhanced well-being under stressful conditions [47–49].

Overall, these findings emphasize the complex interplay between role overload, task distractions, and personal resources in shaping the mental health of healthcare professionals. While individual resilience and emotional intelligence are associated with reduced venerability to stress, broader organizational reforms addressing workload management and interruption control remain essential for supporting sustainable well-being in healthcare systems.

## Novelty and contribution

This study makes several novel contributions. It is one of the first large-scale, multi-site investigations of role overload and psychological outcomes among healthcare workers in Pakistan, a context where such evidence is scarce. Unlike prior studies that examined resilience or emotional intelligence independently, this study integrates both constructs into a single moderated-mediation framework, providing a more comprehensive understanding of how personal resources interact with occupational stressors. By linking these findings to practical recommendations, the study offers context-specific yet

transferable insights for healthcare organizations seeking to strengthen workforce well-being in resource-constrained systems.

## Theoretical implications

The results support the buffering hypothesis, suggesting that personal resources such as resilience and emotional intelligence can mitigate the adverse associations between workload, distractions, and psychological outcomes. Higher resilience was linked with weaker associations between role overload and task distraction, while higher emotional intelligence was associated with weaker links between task distraction and both psychological distress and well-being outcomes. These findings extend stress and coping models by highlighting how individual resources may reduce the impact of occupational stressors in high-demand healthcare settings.

## Practical implications

Hospitals could operationalize resilience and emotional intelligence interventions by embedding structured workshops, mentorship programs, and peer-support groups into existing staff development systems. Protected focus time, digital interruption management tools, and optimized staffing may help reduce task distractions, while resilience-building activities and EI training can enhance coping capacity. Importantly, these interventions are feasible and cost-effective, making them scalable in resource-constrained healthcare systems. Together, such strategies could reduce the psychological toll of workload pressures and contribute to improved workforce well-being and patient care outcomes.

## Limitations and future research

Despite its contributions, this study has several limitations. First, the cross-sectional design restricts the ability to establish causal directionality; future longitudinal and intervention-based research would provide stronger evidence. Second, although validated psychometric scales were employed, the reliance on participant self-reported responses may still introduce common method variance and social desirability bias, which could influence the observed associations. However, the use of widely validated instruments and assurances of confidentiality helped to reduce these risks. Third, the findings are context-specific, reflecting the cultural and institutional realities of Pakistani hospitals. Healthcare systems with stronger institutional support, different staffing structures, or alternative cultural norms may show different patterns, limiting the generalizability of results.

Future research should build on these findings through longitudinal and multi-wave designs, incorporation of observer-rated or objective measures of workload and distraction, and intervention studies. For instance, experimental trials testing resilience training, EI development programs, or digital interruption management tools in real-world hospital settings could provide stronger causal evidence and inform policy-level reforms.

## Conclusions

This study concludes that role overload is a key predictor of psychological distress and reduced well-being among healthcare professionals, with task distraction mediating these effects. Furthermore, employee resilience and emotional intelligence significantly moderate these relationships, highlighting their protective roles. These findings underline the importance of addressing both individual and organizational factors to enhance mental health and job satisfaction in healthcare settings. Practical recommendations include implementing emotional intelligence training, workload management strategies, distraction-reduction measures, and resilience-building programs, along with promoting work-life balance and providing mental health support. Future research should focus on longitudinal studies, cross-cultural analyses, and additional moderators such as job satisfaction and organizational culture, to further guide effective intervention strategies

## Supporting information

**S1 Data. Raw dataset used for statistical analysis.**
(XLSX)

**S1 Questionnaire. Survey tool/Research instrument.**
(DOCX)

## Acknowledgments

The authors would like to express sincere gratitude to Dr. Sumbal Shahbaz for her invaluable supervision, guidance, and mentorship throughout the research process. Special thanks to the faculty of the University of Lahore for their academic support and to all healthcare professionals who participated in this study and shared their valuable experiences. My heartfelt appreciation goes to my parents for their unwavering support and encouragement during my academic journey.

## Author contributions

**Conceptualization:** Faseeh Iqbal.

**Data curation:** Faseeh Iqbal.

**Formal analysis:** Faseeh Iqbal, Fatima Noreen.

**Investigation:** Faseeh Iqbal.

**Methodology:** Faseeh Iqbal, Fatima Noreen, Sami Iqbal.

**Project administration:** Fatima Noreen.

**Resources:** Fatima Noreen, Sami Iqbal, Umer Farooq.

**Software:** Sami Iqbal, Umer Farooq.

**Supervision:** Sumbal shahbaz.

**Validation:** Sumbal shahbaz.

**Writing – review & editing:** Quratulain Batool, Sumbal shahbaz.

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
