## [Decision Letter · Decision Letter 0]

22 Aug 2025

PMEN-D-25-00293

The Impact of Role Overload on Healthcare Workers’ Psychological Distress and Well-Being: The Mediating Role of Task Distraction and Moderating Roles of Resilience and Emotional Intelligence

PLOS Mental Health

Dear Dr. shahbaz,

Thank you for submitting your manuscript to PLOS Mental Health. After careful consideration, we feel that it has merit but does not fully meet PLOS Mental Health’s publication criteria as it currently stands. Therefore, we invite you to submit a revised version of the manuscript that addresses the points raised during the review process.

We look forward to receiving your revised manuscript.

Kind regards,

Lambert Zixin Li, Ph.D.

Academic Editor

PLOS Mental Health

Journal Requirements:

1. We have amended your Competing Interest statement to comply with journal style. We kindly ask that you double check the statement and let us know if anything is incorrect.

2. We note that your Data Availability Statement is currently as follows: “The data is available within the manuscript”

Additional Editor Comments (if provided):

Thank you for submitting your manuscript to PLOS Mental Health. Please carefully address the reviewers’ comments in your revision. In addition, we ask that you address the following points:

Regression analyses: In tables, please report p-values as “<0.001” rather than “0.001.”

Table 6: Hypotheses cannot be “accepted”; only the null hypothesis may be rejected. Please revise the language for statistical accuracy.

Methods section: Ensure in-text citations are accurate. In particular, when citing Role Conflict, Ambiguity, and Overload: A 21-Nation Study, please use the title exactly as published and do not alter it.

Causal vs. associational language: Please replace causal terms (e.g., affect, influence, increase) with associational terms (e.g., related to, covary with, associated with) to reflect the cross-sectional nature of your design.

Language and style: Please have a copy editor review the manuscript for English usage. For example, “this study utilizes” should begin a new sentence rather than follow a comma.

We look forward to receiving your revised manuscript.

Reviewers' comments:

Reviewer's Responses to Questions

**Comments to the Author**

1. Does this manuscript meet PLOS Mental Health’s publication criteria?

Reviewer #1: Yes

Reviewer #2: Yes

2. Has the statistical analysis been performed appropriately and rigorously?

Reviewer #1: Yes

Reviewer #2: Yes

3. Have the authors made all data underlying the findings in their manuscript fully available (please refer to the Data Availability Statement at the start of the manuscript PDF file)?

Reviewer #1: Yes

Reviewer #2: Yes

4. Is the manuscript presented in an intelligible fashion and written in standard English?

Reviewer #1: Yes

Reviewer #2: Yes

Reviewer #1: Dear Dr. Shahbaz and Co-authors,

Thank you for submitting your manuscript titled “The Impact of Role Overload on Healthcare Workers’ Psychological Distress and Well-Being: The Mediating Role of Task Distraction and Moderating Roles of Resilience and Emotional Intelligence” to PLOS Mental Health. Your study addresses a critical issue in healthcare workforce well-being and contributes valuable insights into the psychological dynamics affecting healthcare professionals in high-pressure environments.

After a thorough review, I commend the following strengths of your work:

Strengths

• Relevance and Timeliness: The topic is highly pertinent, especially in the post-pandemic context, where healthcare worker burnout and mental health challenges are increasingly recognized.

• Sample and Scope: The inclusion of 600 healthcare professionals across multiple cities and roles enhances the generalizability of your findings.

• Methodological Rigor: Use of validated scales and Hayes’ PROCESS macro for mediation and moderation analysis demonstrates strong analytical design.

• Ethical Compliance: Ethical approval and informed consent procedures are clearly documented.

However, to meet the publication standards of PLOS Mental Health, the manuscript requires major revisions. See the attached

Reviewer #2: The study highlights an important issue, however, after a careful reading I have noted the following points.

Need of the study should be revised.

discussion section should be written in detail.

rethink on limitations of the study and FUTURE RECOMMENDATIONS

**Do you want your identity to be public for this peer review?** For information about this choice, including consent withdrawal, please see our Privacy Policy

Reviewer #1: **Yes: ** Dr David Onchonga

Reviewer #2: No

---

## [Editor Report · Decision Letter 1]

4 Nov 2025

The Impact of Role Overload on Healthcare Workers’ Psychological Distress and Well-Being: The Mediating Role of Task Distraction and Moderating Roles of Resilience and Emotional Intelligence

PMEN-D-25-00293R1

Dear Dr shahbaz,

We are pleased to inform you that your manuscript 'The Impact of Role Overload on Healthcare Workers’ Psychological Distress and Well-Being: The Mediating Role of Task Distraction and Moderating Roles of Resilience and Emotional Intelligence' has been provisionally accepted for publication in PLOS Mental Health.

Best regards,

Lambert Zixin Li, Ph.D.

Academic Editor

PLOS Mental Health

Dear authors,

Thank you for revising your manuscript and responding carefully to the reviewers’ comments. I have reviewed your revised paper and response memo, and find the revisions satisfactory. The manuscript is now of publishable quality. Congratulations, and thank you for choosing our journal for your work.

Kind regards,

Lambert Zixin Li, PhD